# Experience and Future Perceived Risk of Floods and Diarrheal Disease in Urban Poor Communities in Accra, Ghana

**DOI:** 10.3390/ijerph15122830

**Published:** 2018-12-12

**Authors:** Mumuni Abu, Samuel Nii Ardey Codjoe

**Affiliations:** Regional Institute for Population Studies, University of Ghana, Legon, Accra GA521, Ghana; scodjoe@ug.edu.gh

**Keywords:** flooding, diarrheal disease, risk perception, urban poor, Accra

## Abstract

Diarrheal disease is a critical health condition in urban areas of developing countries due to increasing urbanization and its associated problems of sanitation and poor access to good drinking water. Increasing floods in cities have been linked to the risk of diarrheal disease. There are few studies that specifically link flooding with diarrhea diseases. This may be due to the fact that secondary data mainly hospital recorded cases, and not individual cases at the household level are used. Furthermore, of the few papers that consider the flood-diarrheal diseases nexus, none have considered risk perceptions in general, and more specifically, whether households that have experienced floods which resulted in a reported case of diarrhea, have higher perceived risks of future occurrences of the two phenomena compared to households that had different experiences. Yet, this is critical for the development of interventions that seek to increase protective behaviors and reduce the risk of contracting diarrhea. We surveyed 401 households in some selected urban poor communities in Accra, the capital of Ghana. Results show that households that experienced floods which resulted in a reported case of diarrhea, have higher perceived risk of future occurrence of the two phenomena compared to other households. We recommend public education that reduces the risk of exposure to flood and diarrhea through flood mitigation measures, including the construction of drains in communities and educating communities on good sanitation.

## 1. Introduction

Diarrheal diseases have become a critical health condition in developing countries due to increasing urbanization and its associated problems of sanitation, and poor access to good drinking water. Poor water and sanitation contribute to approximately 94 percent of the four billion cases of diarrhea that occur globally each year [1,2,3,4]. Further, diarrhea is responsible for the death of over two million people annually, representing four percent of total worldwide mortality. In addition, diarrhea causes 1.3 million deaths in children annually [5]. Studies have shown that children younger than five years in developing countries have a median of three episodes of diarrhea annually [6]. The disease is also known to mostly affect poverty-stricken populations [7]. In Ghana, the media have reported impacts of flooding on diarrheal disease incidence in the city of Accra, and there have been warnings from health professionals about the possible outbreak of cholera and non-cholera diarrheal disease after flood disasters. Furthermore, reports from the Ministry of Health of Ghana, indicates that diarrheal disease is among the top ten diseases at the out-patient department of the study district [8]. Even though the development of the oral rehydration solution in the 20th century has served as a remedy for more than 90 percent of dehydration from diarrhea, it has not reduced diarrhea incidence [9].

Diarrheal diseases, including cholera (*Vibrio cholerae*), are transmitted through drinking water or environmental exposure to seawater and sea food [10]. The risk of infection has been linked to temperature and rainfall and related climatic events including the ENSO phenomenon [11,12,13,14,15], sea surface temperature [16], sea and river levels, sea chlorophyll, and plankton [17]. There are few studies that specifically link flooding with diarrhea diseases [18,19,20,21,22]. This may be due to the fact that secondary data, which are mainly hospital-recorded cases, and not individual cases at the household level are often used. For example, there have been confirmed laboratory cases of diarrhea during and after floods in Dhaka, Bangladesh [4,18]. Furthermore, of the few papers that consider the flood-diarrheal diseases nexus, none have considered risk perceptions in general, and more specifically, whether households that have experienced floods which resulted in a reported case of diarrhea have higher perceived risks of future occurrences of the two phenomena compared to households that had different experiences. This is critical for the development of interventions that seek to increase protective behaviors and reduce the risk of contracting diarrhea.

## 2. Risk Perceptions

Risk perceptions have become an important topic to policy makers concerned with risk management and safety issues [23]. There are several schools of thought on the study of risk perception. These can broadly be grouped into three, namely, ecological, psychological, and epidemiological [24,25,26]. Various disciplines under each school of thought share a common idea and so most risk perception studies have become interdisciplinary. Most psychological studies in this area were developed around the theory of affect [25], while other geographical and epidemiological studies in the area were conceptualized around the vulnerability theory [27,28,29]. In both situations, the goal was to identify the population at risk for appropriate policy interventions.

Risk perceptions are known to influence the way people react to situations [8,24,29]. The perceptions people have about a disease informs the kind of measures they will employ to address it. Perceived health risk is critical to individual decision making and important for public health policy formulation. Examining households perceived risk of diarrhea due to previous experience of flood and diarrhea is critical in an urban poor context where there is limited data to undertake community level analysis [20,30,31] and an increasing level of vulnerability to flood risk [32,33].

## 3. Why Focus on Flood-Prone Urban Poor Communities?

Urban settlements south of the Sahara have a high risk of instability due to their haphazard nature, which creates unpredictable changes in the environment that can fuel disease outbreaks. In Ghana, urban settlements are vulnerable to a number of environmental hazards and diseases due to the unplanned nature of settlements and inadequate infrastructure provision [32,34,35]. There is limited access to sanitation facilities and pipe-borne water in the dwellings of the majority of the population residing in urban poor communities [4,30]. In addition, due to the densely populated setting, there is the problem of rapid accumulation of waste in these settlements, and this creates an atmosphere conducive for the transmission of infectious diseases [32,36]. In addition to all these is the increased paving of open spaces and increased water run-off in an environment where drainage systems are often clogged by waste materials [37,38]. This situation provides good grounds for flooding and health related problems in urban poor communities.

To exacerbate the conditions discussed above, there is limited government support for these communities because some of them are classified as informal settlements. However, the population in urban poor communities is growing faster than those in middle and high-class communities in urban areas in Ghana [33]. The cost of accommodation is inexpensive and so most migrants moving into the city settle in these communities. There are equally a lot of informal economic activities in these areas which serve as a source of employment [39]. The population density of urban poor communities in Accra exceeds 25,000 persons per km^2^ compared to an overall average of 6930 persons per km^2^ in the Accra metropolitan area [40]. Flooding is used as the main hazard in this paper because of its high frequency in the study areas and a major environmental problem globally [33].

### Study Area

The two study communities, namely, James Town and Agbogbloshie are located in flood prone areas of the Accra Metropolitan Area [31,41]. These communities are known to have frequent outbreaks of cholera, and a high incidence of non-cholera diarrhea and other diseases in the city [22]. Furthermore, they are communities with low-income and a high proportion of children under five years who may have weak immunity to the frequent environmental problems. These two communities present different environmental contexts of urban poor communities in Accra which are conducive for the study. James Town is generally a paved community, and the housing structures are constructed with concrete blocks and mud bricks, while Agbogbloshie is marshy with few concrete blocks and majority make-shift structures [41]. Both communities are classified as slum and located at different points of the Odaw River which serves as the major drainage system for the Accra metropolitan area (Figure 1). The Odaw River flows through the Agbogbloshie community and drains into the sea at James Town.

## 4. Methodology

Data for the study is from a household survey conducted in the two study communities, namely, James Town and Agbogbloshie. The survey was part of the Regional Institute for Population Studies’ (RIPS) EDULINK Round II survey. EDULINK is an urban health and poverty project of RIPS, University of Ghana. The aim of the project is to link research to communities. The study was approved by the Noguchi Memorial Institute for Medical Research Institutional Review Board (NMIMR-IRB) of the University of Ghana. Approval for the study was given on 14 November 2012 with study number NMIMR-IRB CPN 041/12-13. Data collection was done for a period of six weeks, i.e., from 16 November 2012 to 31 December 2012. All study respondents signed, or thumb printed an informed consent form that was approved by the NMIMR-IRB before participating in the study. The reference period of interest in the study was the first four weeks following the 26 October 2011 flooding event in Accra.

A total of eight enumeration areas (EAs) from James Town and five from Agbogbloshie were selected using simple random sampling technique. There was more sampling in the EAs in Agbogbloshie as compared to that of James Town because of the larger number of EAs in James Town. Approximately 40 households were targeted for each of the EAs in Agbogbloshie, and so a total of 200 households were sampled. In James Town on the other hand, a total of 30 households were targeted for each EA, and a total of 240 households were sampled. In all, 199 and 202 households were interviewed in Agbogbloshie and James Town, respectively, totaling 401 households in both communities. The response rate was approximately 91 percent.

Structured questionnaires were used to solicit information on the general characteristics of households including age and sex of the members of households, household size, educational level, assets, experience of flooding and diarrhea in the past 12 months preceding the survey, and when exactly diarrhea was diagnosed. Finally, data on risk perceptions about diarrheal disease resulting from flooding, source of drinking water, sanitation and hygiene conditions of households were collected. These variables are critical in examining the health risk of a population to a hazard [8,33].

### 4.1. Dependent Variable

The dependent variable is a numerical risk-measure which considers perceived risk of future diarrhea infection as a result of previous experience with flood which resulted in diarrhea. This measure provides an opportunity to ascertain level of risk from populations with different backgrounds. The data collection team provided an explanation to each participating household on how to score their risk using 10 beans. Each bean selected represented 10 points, and thus, 10 beans equals 100%. This method was used to bring all the participants to the same numeracy level to be able to provide accurate estimates in percentages. Household heads were asked to numerically rate the probability (on a scale of 0% to 100%) of any household member being infected with diarrhea because of future flood event. A score of 0% is interpreted as no chance of diarrheal disease infection, while 100% indicates a definite infection of diarrhea.

### 4.2. Independent Variable

The independent variable is household experience of flooding which resulted in a reported case of diarrhea. We used two questions to create this variable. First, whether the household experienced the 26 October 2011 flooding event, and, second, whether any member of the household reported diarrhea within four weeks after the flooding event. Households affected by the flooding event are defined as households who were in the community on 26 October 2011 when the flooding event occurred, and their household was directly affected. To distinguish between households that were affected and those not affected, information on the year and the month in which households were affected was collected. Households that were not affected are generally located in elevated areas in the community, while those affected are mainly located close to the lagoon and sea, along main storm drains or reside in low lying areas of the communities. Thus, the independent variable is categorized as follows: 1 = Household experienced flood and had at least a reported case of diarrhea; 2 = Household experienced flood but had no reported case of diarrhea; 3 = Household did not experience flood but had at least a reported case of diarrhea; and 4 = Household did not experience flood and had no reported case of diarrhea. We hypothesize that households that experienced floods which resulted in a reported case of diarrhea, are more likely to have a higher perceived risk of future occurrences of the two phenomena compared to households that had different experiences.

### 4.3. Control Variables

To examine the true strength of the independent variable on perceived risk of diarrheal disease, we controlled for the effect of other variables. These include socio-demographic characteristics of household, household diarrhea preventive strategies, water and sanitation practices, and environmental risk factors. Household socio-demographic characteristics, including mean age of household members, proportion of household members with some formal education, sex of household head, wealth status and household size, are factors that have been reported in the literature to affect diarrhea incidence [4,42].

Household diarrhea preventive measures such as washing of hands with soap before eating and after visiting the toilet are examined. There have been several ‘hand wash’ programs in developing countries aimed at increasing awareness towards good hygiene practices [43]. In addition, the presence of livestock including sheep and goats, and the number of times cockroaches have been sighted in the household in the past seven days preceding the survey, are also used to assess household sanitary conditions. Livestock and cockroaches are known to carry the bacteria that cause diarrhea and their presence in the household could lead to contamination of household water and food and subsequently a diarrheal disease incidence [43,44].

Epidemiologically, it has been established that environmental risk factors such as distance to nearest public toilet and refuse collection point, are significant predictors of diarrheal disease [37,44].

Furthermore, household water and sanitation are critical factors in examining diarrhea incidence [45,46,47]. Thus, type of toilet facility, mode of disposing solid waste and source of drinking water are included in the models. Mode of solid waste disposal of the households is categorized as improved and unimproved methods based on World Health Organization classifications [48]. The improved sources of disposing solid waste are: availability of refuse bins or containers collected regularly by public or private companies and the unimproved means are using the services of truck pushers (*kaya bola*), and indiscriminate disposal of waste especially into community drains. The details of the variables coding are explained in Table 1.

### 4.4. Analytic Approach

The survey data was analyzed using descriptive statistics and a bivariate analysis for the variables of interest in the study. In addition, a multivariate analysis of the independent and the dependent variable was examined while controlling for socio-demographic and other environmental factors. This was to establish the statistical association between the independent variables and the dependent variable. In all, two models were fitted. The first model examined the association between household previous experience of flooding and diarrheal disease, and their perceived risk of diarrheal disease. We did not consider any other variables at this stage because we wanted to know the statistical association between the main independent variable and the dependent variable without the influence of any other variable. To test the robustness of the association between experience of flooding and diarrheal disease and perceived risk of diarrhea infection, we controlled for the effect of other variables in model 2. The other variables include socio-demographic characteristics of household members, household water and sanitation, and environmental risk factors. The level of significance for interpreting the results is set at *p* < 0.05.

## 5. Results

### 5.1. Descriptive Statistics of Outcome, Explanatory and Control Variables

Table 2 shows that more than one-quarter (30.0%) of households in the study communities (52.3% in Agbogbloshie and 7.4% in James Town) experienced both the 26 October 2011 flooding and had members reporting of diarrhea within four weeks after the flood. Also, 20% of the households in the communities (13% in Agbogbloshie and 26% in James Town) did not experience the flood but reported members experiencing diarrhea within four weeks after the flood.

The experience of both flooding and diarrhea in the study communities is expected to trigger some perceptions among households in the study communities. Numerically, we measured households’ perceived risk of diarrheal disease as a result of previous exposure to flooding and diarrhea on a scale of 0–100%.

As shown in Table 2, the mean perceived risk of diarrheal disease as a result of previous exposure to flooding is 18% in Agbogbloshie compared to about 5% in James Town. Thus, households in Agbogbloshie have higher perceived risk of diarrheal disease than those in James Town. The lower mean perceived risk score in James Town compared to Agbogbloshie could be attributed to the numerous sanitation interventions in the area including the distribution of refuse bins and the construction of ally pavements to facilitate free flow of rain water. The Agbogbloshie community has received less attention from government and other organizations in terms of sanitation facilities and drainage construction because of legal contestations on the land for residential purposes. In addition, Agbogbloshie is more prone to flooding as compared to James Town.

Table 2 shows that there are more males in Agbogbloshie (58.8%) compared to James Town (54.5%) whilst the mean age of household members is higher in James Town (32.7 years) compared to Agbogbloshie (25.2 years). The proportion of households with all members with some form of education is slightly higher in James Town (71.8%) compared to Agbogbloshie (70.9%) and the mean household size is almost the same in both localities (i.e., 3.3 in James Town and 3.2 in Agbogbloshie). In terms of the housing condition of the study communities, the common material used for the floor of the houses in both communities are sand/cement/concrete (84% in Agbogbloshie and 73% in James Town) whilst the main material used for the wall of the house is wood (56% in Agbogbloshie and 45% in James Town). There is an equal distribution of wealth across the wealth status for the two study communities together.

Table 2 further shows that there are more households in James Town than in Agbogbloshie with members with the following characteristics: hand washing with soap before eating (28.2% vs. 23.1%); have livestock in household (3.7% vs. 2.0%). However, there are more households in Agbogbloshie (43.2%) compared with James Town (35.1%) with members that practice hand washing with soap after visiting the toilet. The frequency of sighting of cockroaches in households is much higher in Agbogbloshie (63.3% for 4 times or more in the past seven days preceding the survey) compared to 47.4% in James Town. In addition, while a higher proportion of households live closer to public toilets in Agbogbloshie (35%), majority of the households in James Town (99%) live closer to refuse collection points. Public toilet is the most common toilet facility used in both communities (89.4% in Agbogbloshie and 80.2% in James Town). Also, the use of improved solid waste disposal facilities is higher in James Town (82.7%) compared to Agbogbloshie (59.3%). Finally, the use of piped water (into dwelling, yard and in public) for drinking is higher in James Town (44.5%) compared to Agbogbloshie (26.1%).

### 5.2. Factors Associated with Perceived Risk of Diarrheal Disease

Results of the bivariate analysis presented in Table 3 show that households experience of flooding event and diarrhea, sex of household head, mean age of household members, the main material on the wall of the house, main source of drinking water, type of toilet facility, number of times cockroaches are sighted in household, distance to the nearest refuse collection point, and locality of residence are the variables with statistically significant associations with households perceived risk of diarrheal disease.

### 5.3. Predictors of Perceived Risk of Diarrheal Disease

Model 1 in Table 4 shows that without controlling for the effect of socio-demographic factors, household water and sanitation, and environmental risk factors, the experience of both the flooding event and a member of a household experiencing diarrhea within four weeks after the flood are significant predictors of perceived risk of diarrheal disease in the study communities. Households that experienced the flood but did not report diarrhea, those that did not experience the flood but reported diarrhea, and those that did not experience the flood and also did not report diarrhea had less perceived future risk of diarrheal disease compared with those who experienced the flood and reported diarrhea incidence.

In model 2 in Table 4, the effects of the socio-demographic and environmental factors were controlled to test the robustness of the effects of experience of flooding and diarrheal disease on household perceived risk of diarrheal disease. The results show that experience of flood and diarrheal disease, sex of household head, main materials on the wall of the house, and locality of residence are significant predictors of households perceived risk of diarrheal disease. Whilst the experience of both flood and diarrhea, sex of household head and locality of residence had a negative effect on perceived risk of diarrheal disease, the material used for the wall of the house had a positive effect. The locality of residence of the individual households is a very significant predictor, which also support the bivariate analyses that indicates that the two localities are very different socio-economically and environmentally.

Model 2 in Table 4 also show that households that experience both the flood event and diarrhea have a higher risk of diarrheal disease compared with the other households. Also, female-headed households have 2.8 less perceived risk of diarrheal disease compared to male-headed households. Households that have their walls made of plywood have higher perceived risk of diarrheal disease compared with those whose walls are made of bamboo with mud. Finally, because of the differences in the localities, households in James Town have less perceived risk of diarrheal disease compared to those in Agbogbloshie [49].

## 6. Discussion

The study hypothesis was that households that previously experienced flooding which resulted in diarrhea incidence will have higher risk perceptions of future occurrence of the two phenomena because evidence in literature show a strong correlation between flooding and diarrheal disease [11,19,36]. Our results confirm that after controlling for all socio-demographic and environmental factors, households that experienced the 26 October 2011 flood and reported diarrheal disease among members within four weeks after the flood have a higher perceived risk of developing diarrhea following future floods than other households. The results from the analysis provides us a better understanding of factors that influence households perceived risk of diarrheal diseases in urban poor communities for public health interventions.

At the bivariate analyses, we found that the experience of both the flood event and diarrhea, sex of household head, mean age of household members, main material on the wall of house, main source of drinking water, type of toilet facility, the number of times seen cockroaches at home in the past seven days preceding the survey, distance from home to the nearest refuse collection point and locality of residence had significant associations with perceived risk of diarrheal disease. These findings are similar to other descriptive studies that examined the association between flooding and diarrheal disease [44,50]. Finally, at the multivariate analysis whilst controlling for all other variables, the experience of both the flood event and diarrhea, sex of household head, material used for the wall of house and locality of residence are significant predictors of perceived risk of diarrheal disease following exposure to future floods.

A higher proportion of the households in Agbogbloshie were exposed to the 26 October 2011, flooding, experienced diarrheal disease within four weeks after the flooding and they generally have a higher perceived risk of diarrheal disease. Perceptions are critical in all actions that are taken by human population. Apart from households that experienced both the flood event and diarrhea, all other households had less perceived risk of diarrhea following future flood. The low perceived risk of diarrhea among the other households is not good for public health education. This is because households that reside in high-risk localities and do not see themselves at risk will not do anything to prevent the occurrence of the event. It is crucial to develop interventions that educate the population about the general vulnerability of their community, which has an effect on them and not to only concentrate on their immediate household.

## 7. Conclusions

In this paper, we used household surveys in selected low-income areas in Accra, the capital of Ghana, to examine the effect of previous experience of both flooding and diarrhea on households perceived risk of diarrheal disease following future flood. This is because the risk perceptions people develop because of their exposure to an environmental hazard influences the strategies they will employ to avert its effects.

The perception among the study population is that household experience of flooding and diarrhea is a significant predictor of household perceive risk of diarrhea following future flood. Households that experienced the flood event and had members reporting diarrhea within four weeks after the flood have higher perceived risk of diarrhea than other households. The lower perceived risk levels by the other household in a very vulnerable environment is an indication of how they will respond to interventions that are aimed at reducing population risk to diarrhea following future floods. The two study localities present different socio-economic and environmental context of an urban poor area. Whilst James Town locality presents good sanitation and socio-economic environment, the Agbogbloshie community has poor environmental condition and higher incidence of diarrheal disease.

Our main recommendation is that programs aimed at addressing the health effects of floods should involve community members from the beginning so that they can associate themselves with the final outcomes. Involving community members requires a critical understanding of how they perceive and associate themselves with the risk of the issues being discussed. In addressing most global situations, risk perceptions are usually the last option policy implementers consider because of the difficulty in evaluating risk perceptions. However, our actions and inactions are generally influenced by our perceptions and it is important to make it a priority in decision making. Understanding household perceived risk to diarrhea following future flood helps to identify households that perceive themselves at high or low risk of diarrheal disease. The perceived risk of the households also determines the kind of measures they will put in place to protect members from the impact of future flood. Households that have higher perceived risk are more likely to develop positive attitudes towards ensuring that they are not infected with diarrhea in the event of flood compared to other households.

The measure of diarrheal disease was self-reported and not from medical diagnoses which could compromise the understanding of the respondents as to what is meant by diarrheal disease. There were detailed explanations of what diarrheal disease is in the local language to minimize errors. Despite this limitation, the robustness of the methods used provides some lessons for researchers interested in similar studies in urban poor communities across the globe.

## Figures and Tables

**Figure 1 ijerph-15-02830-f001:**
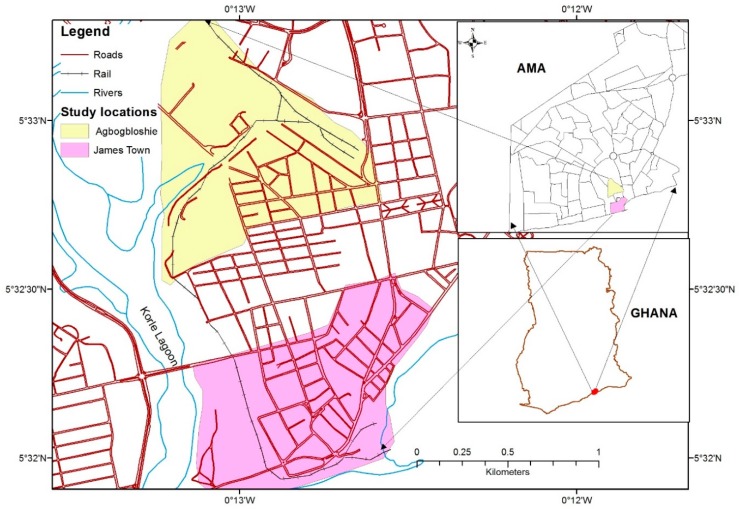
Map of study area.

**Table 1 ijerph-15-02830-t001:** Description of control variables.

**Variables and Their Classification**	**Coding**
Mean age of Household Members	Continuous variable
Household size	
**Sex of household head**	
Male	=1 Male, =2 Female
Female	
**Proportion of household members with some education**	
No member had education	=1 No member had education, =2 Less than 50% had education, =3 50% and above had education, =4 All members had education
Less than 50% had education	
50% and above had education	
All members had education	
**Wealth status**	
Poorest	=1 Poorest, =2 Poor, =3 Middle, =4 Rich, =5 Richest
Poor	
Middle	
Rich	
Richest	
**Hand washing with soap before eating**	
Yes	=1 Yes, =2 No
No	
**Hand washing with soap after visiting the toilet**	
Yes	=1 Yes, =2 No
No	
**Main material on the floor of the house**	
Sand/Cement/Concrete	=1 Sand/Cement/Concrete, =2 Wood/Wood Planks/Woolen Carpet, =3 Ceramic Tiles/Porcelain Granite/Marble
Wood/Wood Planks/Woolen Carpet	
Ceramic Tiles/Porcelain Granite/Marble	
**Main material on the wall of the house**	
Bamboo with mud	=1 Bamboo with mud, =2 Wood, =3 Plywood, =4 Cement Blocks/Concrete, =5 Other
Wood	
Plywood	
Cement Blocks/Concrete	
Other	
**Main source of drinking water**	
Piped into dwelling	=1 Piped into dwelling, =2 Piped into yard, =3 Public tap/stand pipe, =4 Sachet water/bottled water
Piped into yard	
Public tap/stand pipe	
Sachet water/bottled water	
**Type of toilet facility**	
No facility/bucket pan/pit latrine	=1 No facility/bucket pan/pit latrine, =2 WC/Flush toilet, =3 KVIP, =4 Public toilet
WC/Flush toilet	
Kumasi Ventilated-Improved Pit (KVIP)	
Public toilet	
**Mode of disposing solid waste**	
Improved	=1 Improved, =2 Unimproved
Unimproved	
**Used soap to wash hands before eating**	
Yes	=1 Yes, =2 No
No	
**Used soap to wash hands after visiting toilet**	
Yes	=1 Yes, =2 No
No	
**Availability of livestock at home**	
Yes	=1 Yes, =2 No
No	
**Number of times seen cockroaches at home in the past 7 days**	
Never	=1 Never, =2 1–3 Times, =3 4 or more Times, =4 Don’t Know
1–3 Times	
4 or more Times	
Don’t Know	
**Distance from home to the nearest public toilet**	
Less than 50 m	=1 Less than 50 m, =2 50 m and above
50 m and above	
**Distance to the nearest refuse collection point**	
Less than 50 m	=1 Less than 50 m, =2 50 m and above
50 m and above	
**Locality**	
Agbogbloshie	=1 Agbogbloshie, =2 James Town
James Town	

**Table 2 ijerph-15-02830-t002:** Description of household perceived risk of diarrhea after previous experience of flood and diarrhea, socio-demographics, economic and water and sanitation of households.

Variable	Agbogbloshie		James Town		Total	
	Count (Mean)	% (SD)	Count (Mean)	% (SD)	Count (Mean)	% (SD)
**Outcome**						
Household perceived future risk of diarrhea	(18.14)	(22.16)	(4.48)	(6.62)	(11.26)	(17.67)
**Explanatory Variables**						
Experience of 26 October 2011 flooding and experience of diarrhea within 4 weeks after the flood						
Household experienced flood and had at least a reported case of diarrhea	104	52.3	15	7.4	119	29.7
Household experienced flood but had no reported case of diarrhea	46	23.1	17	8.4	63	15.7
Household did not experience flood but had at least a reported case of diarrhea	26	13.1	52	25.7	78	19.5
Household did not experience flood and had no reported case of diarrhea	23	11.6	118	58..4	141	35.2
Sex						
Male	117	58.8	110	54.5	227	56.6
Female	82	41.2	92	45.5	174	43.4
Mean age of household members	(25.20)	(8.76)	(32.67)	(14.45)	(28.96)	(12.53)
Education of household members						
Less than 50% had education	5	2.5	6	3.0	11	2.7
50% and more had education	53	26.6	51	25.2	104	25.9
All members had education	141	70.9	145	71.8	286	71.3
Household size	(2.22)	(0.85)	(2.19)	(0.99)	(2.2)	(0.926)
Main material on floor of house						
Sand/Cement/Concrete	167	83.9	147	72.8	314	78.3
Wood/Wood Planks/Woolen Carpet	25	12.6	47	23.3	72	18.0
Ceramic Tiles/Porcelain Granite/Marble	7	3.5	8	4.0	15	3.7
Main material on wall of house						
Bamboo with mud	14	7.0	7	3.5	21	5.2
Wood	112	56.3	91	45.0	2013	50.6
Plywood	30	15.1	23	11.4	53	13.2
Cement Blocks/Concrete	39	19.6	75	37.1	114	28.4
Other	4	2.0	6	3.0	10	2.5
Wealth status						
Poorest	20	10.1	60	29.7	80	20.0
Poor	11	5.5	69	34.2	80	20.0
Middle	46	23.1	35	17.3	81	20.2
Rich	61	30.7	19	9.4	80	20.0
Richest	61	30.7	19	9.4	80	20.0
Main source of drinking water						
Piped into dwelling	1	0.5	22	10.9	23	5.7
Piped into yard	6	3.0	17	8.4	23	5.7
Public tap/stand pipe	45	22.6	51	25.2	96	23.9
Sachet water/bottled water	147	73.9	112	55.4	259	64.6
Type of toilet facility						
No facility/bucket pan/pit latrine	1	0.5	10	5.0	11	2.7
WC/Flush toilet	0	0.0	20	9.9	20	5.7
Kumasi Ventilated-Improved Pit (KVIP)	20	10.1	10	5.0	30	7.5
Public toilet	178	89.4	162	80.2	340	84.8
Mode of disposing solid waste						
Improved	118	59.3	167	82.7	285	71.1
Unimproved	81	40.7	35	17.3	116	28.9
Used soap to wash hands before eating						
Yes	46	23.1	57	28.2	103	25.7
No	153	76.9	145	71.8	298	74.3
Used soap to wash hands after visiting toilet						
Yes	86	43.2	71	35.1	157	39.2
No	113	56.8	131	64.9	244	60.8
Availability of livestock at home						
Yes	4	2.0	11	5.4	15	3.7
No	195	98.0	191	94.6	386	96.3
Number of times seen cockroaches at home in the past 7 days						
Never	36	18.1	53	26.2	89	22.2
1–3 Times	36	18.1	76	37.6	112	27.9
4 or more Times	126	63.3	64	31.7	190	47.4
Don’t Know	1	.5	9	4.5	10	2.5
Distance from home to the nearest public toilet						
Less than 50 m	69	34.7	64	31.7	133	33.2
50 m and above	130	65.3	138	68.3	268	66.8
Distance to the nearest refuse collection point						
Less than 50 m	105	52.8	199	98.5	304	75.8
50 m and above	94	47.2	3	1.5	97	24.2
N	199		202		401	

**Table 3 ijerph-15-02830-t003:** Association between socio-demographics, economic, and water and sanitation of households and household perceived risk of diarrhea as a result of previous experience of flood and diarrhea.

Variable	Mean Perceived Risk of Diarrhea (SD)	F	*p*-Value
Experience of 26 October 2011 flooding and experience of diarrhea within 4 weeks after the flood		44.4	0.001
Household experienced flood and had at least a reported case of diarrhea	22.94 (23.05)		
Household experienced flood but had no reported case of diarrhea	2.54 (7.82)		
Household did not experience flood but had at least a reported case of diarrhea	15.06 (16.64)		
Household did not experience flood and had no reported case of diarrhea	3.19 (6.33)		
Sex		6.965	0.009
Male	13.28 (20.01)		
Female	8.62 (13.63)		
Mean age of household members	r = −0.199		0.001
Education of household members		0.325	0.722
No member/less 50% had education	15.45 (20.67)		
50% and more had education	10.96 (171.98)		
All members had education	11.21 (17.47)		
Household size	r = −0.055		0.276
Main material on floor of house		0.472	0.624
Sand/Cement/Concrete	10.91 (16.24)		
Wood/Wood Planks/Woolen Carpet	13.06 (23.30)		
Ceramic Tiles/Porcelain Granite/Marble	10.00 (15.58)		
Main material on wall of house		10.744	0.001
Bamboo with mud	18.10 (14.36)		
Wood	10.34 (15.84)		
Plywood	23.40 (27.10)		
Cement Blocks/Concrete	5.92 (12.40)		
Other	12.00 (15.59)		
Wealth status		0.688	0.600
Poorest	13.35 (18.65)		
Poor	12.17 (18.69)		
Middle	10.86 (17.17)		
Rich	10.13 (19.38)		
Richest	9.36 (13.61)		
Main source of drinking water		2.93	0.033
Piped into dwelling	2.17 (5.18)		
Piped into yard	6.52 (8.85)		
Public tap/stand pipe	12.29 (17.50)		
Sachet water/bottled water	12.10 (18.74)		
Type of toilet facility		3.69	0.012
No facility/bucket pan/pit latrine	5.45 (6.88)		
WC/Flush toilet	5.75 (7.48)		
Kumasi Ventilated-Improved Pit (KVIP)	20.17 (28.90)		
Public toilet	10.99 (16.78)		
Mode of disposing solid waste		2.134	0.145
Improved	10.44 (15.37)		
Unimproved	13.28 (22.25)		
Used soap to wash hands before eating		0.175	0.676
Yes	10.63 (18.25)		
No	11.48 (17.48)		
Used soap to wash hands after visiting toilet		0.000	0.989
Yes	11.27 (19.86)		
No	11.25 (16.13)		
Availability of livestock at home		1.758	0.186
Yes	5.33 (11.26)		
No	11.49 (17.83)		
Number of times seen cockroaches at home in the past 7 days		8.714	0.001
Never	7.64 (11.36)		
1–3 Times	7.01 (11.79)		
4 or more Times	15.84 (21.80)		
Don’t Know	4.00 (6.99)		
Distance from home to the nearest public toilet		2.400	0.122
Less than 50 m	9.32 (17.02)		
50 m and above	12.22 (17.92)		
Distance to the nearest refuse collection point		43.32	0.001
Less than 50 m	8.14 (14.33)		
50 m and above	21.03 (22.89)		
Locality		70.393	0.001
Agbogbloshie	18.14 (22.16)		
James Town	4.48 (6.62)		

r = correlation coefficient.

**Table 4 ijerph-15-02830-t004:** Ordinary Least Square model of predictors of households perceived risk of diarrhea as a result of previous experience of flood and diarrhea.

			Robust			Robust
	Coefficient		Std. Err.	Coefficient		Std. Err.
Variable	Model 1			Model 2		
Experience of 26 October 2011 flooding and experience of diarrhea within 4 weeks after (RC is Experienced flood and had diarrhea)						
Household experienced flood but had no reported case of diarrhea	−20.402	***	2.331	−17.813	***	2.213
Household did not experience flood but had at least a reported case of diarrhea	−7.877	**	2.830	−1.816		2.836
Household did not experience flood and had no reported case of diarrhea	−19.750	***	2.181	−11.727	***	2.257
Sex (RC is Male)						
Female				−2.791	*	1.392
Mean age of household members				−0.171		0.283
Wealth status (RC is poorest)						
Poor				−0.812		2.230
Middle				−0.522		2.185
Rich				1.006		2.362
Richest				0.467		1.977
Main material on wall of house (RC is Bamboo with mud)						
Wood				−3.483		2.606
Plywood				8.662	**	3.598
Cement Blocks/Concrete				−4.660		2.867
Other				−1.965		3.823
Main source of drinking water (RC is public tap/stand pipe						
Piped into dwelling				1.849		1.571
Piped into yard				0.269		1.896
Sachet water/bottled water				2.059		1.658
Type of toilet facility (RC is public toilet)						
No facility/bucket pan/pit latrine				1.956		1.868
WC/Flush toilet				4.222		2.336
Kumasi Ventilated-Improved Pit (KVIP)				7.443		4.713
Number of times seen cockroaches at home in the past 7 days (RC is never)						
1–3 Times				−0.469		1.481
4 or more Times				1.275		1.595
Don’t Know				1.653		3.120
Distance to the nearest refuse collection point (RC is Less than 50 m)						
50 m and above				2.265		2.661
Locality (RC is Agbogbloshie)						
James Town				−8.600	***	1.980
Constant	22.941	***	2.114	24.26546	***	3.514
R^2^			0.251			0.397
F-Statistic	(3, 397) = 39.18 ***			(24, 376) = 9.45 ***		

* *p* < 0.05; ** *p* < 0.01; *** *p* < 0.001.

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
