# Peer review of "Experience and Future Perceived Risk of Floods and Diarrheal Disease in Urban Poor Communities in Accra, Ghana"

_ijerph, 2018, doi:10.3390/ijerph15122830_

Round 1
Reviewer 1 Report
attached.

Author Response
We are very grateful to the reviewer for the insightful review and comments which will enrich the quality of the paper. We have revised the title of the manuscript, which is now “Experience of future perceived risk of floods and diarrhoeal disease in urban poor communities in Accra, Ghana”.
Attached is our responses to the comments.
Thank you.

Reviewer 2 Report
The perceived risk of diarrhoeal disease in urban poor communities in Accra, Ghana by Abu et al., containsvaluable information and could be used to minimizethe health risk in the future. However, this manuscript needs major revision including English edit requires before submitting it for publication. One of the major concern is that the methods which do not have a sample questioner. That should be submitted as an appendix or supplementary file. In the methodology section, there were variables, that should be describeddifferently. Section 4.3 would be better with a table and less text. Section 4.4. need what analysis using which software and how it was done. Tables in the result section must change, at the moment they are big and looks like raw information.
Finally, the data were collected from 2012 and may not be valuable now.
In the current form, it looks like a report and needs formatting appropriately before sending it again to a reviewer.
Author Response

(The authors gave the same response as above.)

Reviewer 3 Report
The present study was carried out to examine how prior experiences of flood influences future risk of diarrhoea, which is critical for developing public health education on protective behaviours. As a results, authors suggested that public education on the social and environmental factors that influences people’s perceived risk of diarrhoeal disease, because these are the critical factors that will make them develop a positive environmental behaviour.
This research is very interesting.
However, some modifications are required.
For the abstract, please describe the results more concretely.
Please check again about the method of citing documents.
Author Response

(The authors gave the same response as above.)

Round 2
Reviewer 1 Report
The second submission of this manuscript is greatly improved and has addressed most of the major concerned of my first review. Most importantly, the authors included the very interesting bivariate analysis of experience of flooding and experience of diarrhea post-flooding. I have only a few minor concerns that should at least be considered. The authors did not take my advice on creating a parsimonious statistical model 2 but my sense is that the correct results were found none the less. Overall, a much more coherent and forceful presentation.
Minor concerns.
I do not think that Table 1 is needed. I don’t know if I have ever seen a table like this that simply indicates how variables were coded. Interestingly, I notice that locality, James Town and Agbogbloshie, were not coded.
I am still confused about when the flood happened. Was it October 26, 2011 or November 26, 2011. Both dates seem to be used interchangeably.
The top of page 10 has a Yes/No result but no variable. Is this an error?
Author Response
We are very grateful to the reviewer for the insightful review and comments. Our responses to the comments are attached.
Thank you.

Reviewer 2 Report
The manuscript has improved significantly. I am happy with the manuscript.
Author Response

(The authors gave the same response as above.)
